# Normative Muscle Activation Patterns During One and Five Countermovement Jumps

**DOI:** 10.3390/bioengineering12070767

**Published:** 2025-07-16

**Authors:** Anabel Gallego-Pérez, Elisa Benito-Martínez, Beatriz Alonso-Cortés Fradejas

**Affiliations:** 1San Juan de Dios Foundation, 28016 Madrid, Spain; 2Health Sciences Department, San Juan de Dios School of Nursing and Physical Therapy, Comillas Pontifical University, 28350 Madrid, Spain; 3Nursing and Physical Therapy Department, Health Sciences’s Faculty, León University, Ponferrada Campus, 24401 Castilla y León, Spain

**Keywords:** surface electromyography, normative values, landing phase, take-off phase, biomechanics

## Abstract

Studying normative values for muscle activation in the vastus lateralis (VL), vastus medialis (VM), and biceps femoris (BF), as well as the hamstrings/quadriceps (H:Q) ratio during the Countermovement Jump (CMJ). Determine whether there were differences between the CMJ and the trial of 5 consecutive CMJs (5 CMJ) and between the take-off and landing phases. A cross-sectional descriptive study. Thirty-one participants (20 females and 11 males, 22.52 ± 3.295 years, BMI 24.32, weight 58.23 ± 4.32 Surface electromyography has been used to determine muscle activation during the CMJ and 5 CMJ. Muscle activation in the VL, VM, and BF, as well as the hamstrings/quadriceps ratio in take-off and landing phases of the CMJ and 5 CMJ. The results show normative values in the VL, VM, and BF during both the CMJ and 5 CMJ, with the exception of the BF during the landing phase of the 5 CMJ. In conclusion, the activation in the take-off phase of the VM and VL is greater than during the landing phase. The BF shows similar activation in both the take-off and landing phases. The 5 CMJ does not induce greater muscular fatigue than the CMJ.

## 1. Introduction

Technological Surface electromyography (EMG) is a biomechanical tool that detects muscle activation through surface electrodes, allowing both static and dynamic data to be obtained. It provides valuable information for clinical practice [1], enabling the determination of activation sequences across a wide range of exercises [1,2,3,4,5,6,7,8], even helping to identify the onset of fatigue [9].

Several authors have attempted to determine normative parameters of muscle activation for various movements [10,11,12], with differences being found in terms of exercise velocity and external load [13,14].

Differences between muscles, as well as their varying levels of activation during different exercises, have been studied by various authors. For example, Muyor JM et al. assessed the electromyographic activity of the gluteus medius, gluteus maximus, biceps femoris (BF), vastus lateralis (VL), vastus medialis (VM), and rectus femoris in 20 subjects during the performance of several exercises, finding higher activity in the single-leg squat compared to the forward lunge and lateral step-up, and also observing greater activation during the concentric phase of the exercise compared to the eccentric phase [5].

In physical capacity assessment, jump tests are commonly used [15], with Countermovement Jump (CMJ) being the most commonly employed for assessing explosive strength due to its similarity to a natural jump. It has even been used as a determinant in assessing injury risk [7,15,16,17,18,19,20].

Gross, through electromyography (EMG) and dynamometry, assessed how greater activation was generated in the eccentric deceleration phase of the jump, during a trial of 5 consecutive CMJ (5 CMJ) jumps, compared to that generated in simple CMJ [20].

On the other hand, studying the activation ratios between different muscles is one of the most relevant factors for injury prevention. Several authors have found differences in activation between the quadriceps and hamstrings, with a greater difference observed between the VM and BF compared to the VL and BF [21,22,23].

Determining normal muscle activation patterns in sports gestures in terms of magnitude and timing allows us to detect subjects with a high probability of injury or to find the cause of their dysfunction. For instance, greater activation of the gluteus maximus has been shown in women with patellofemoral pain [24], late and short activation of the gluteus medius in women with knee pain during running [25], and greater activation of the tensor fasciae latae in iliotibial band syndrome [26]. Not only can a correlation be seen with altered activation patterns and pathologies or dysfunctions in the lower limb, it is also evident in the upper limbs, as is the case with shoulder pain in amateur athletes with less activation of the lower trapezius [27]. In addition, we can use these alterations in normal muscle activation patterns to focus a more objective treatment in some pathologies, such as anterior cruciate ligament rupture [28,29].

Muscle imbalances expressed as ratios pose a risk factor for certain injuries, with imbalances between the hamstrings and quadriceps being linked to anterior cruciate ligament (ACL) or hamstring tears [30,31]. These ratios can be defined through dynamometers, such as the Hamstring:Quadriceps (H:Q) ratio, which examines the force relationship between the hamstrings and the quadriceps [14,30,31]. If a concentric contraction is used, a traditional ratio can be established [14], or a functional ratio when using eccentric hamstring contractions against concentric quadriceps contractions. The latter is considered more useful due to its similarity to actions in sports practice [30,32]. Additionally, several studies aim to establish the H:Q activation ratio using EMG for various sporting movements [14].

However, EMG faces the challenge of normalizing activation data to allow intra- and inter-subject comparisons. This process can be done in several ways: through the subject’s maximal voluntary isometric contraction (MVC) [14,31,33,34], by comparing the activation of a muscle belly relative to the overall muscle activity, or by dynamically normalizing the signal [14].

Since the CMJ is one of the most commonly used jumps in injury prevention and evaluation and differences are observed when performed as a single jump or with multiple jumps, and, given that the activation ratio between muscles is another essential tool for preventing injuries in specific sports gestures, it seems crucial to establish normative values for muscle activation, both during the take-off and landing phases of these jumps.

Therefore, the hypothesis of this study proposed the feasibility of determining normative muscle activation values for the vastus medialis (VM), vastus lateralis (VL), and biceps femoris (BF), as well as the hamstring-to-quadriceps (H/Q) ratio during the countermovement jump (CMJ) and a sequence of five consecutive CMJs (5 CMJ). It was further hypothesized that BF activation would be greater during the landing phase in order to support joint stability and that differences would be observed between the take-off and landing phases of each jump, with greater quadriceps activation during take-off and greater biceps femoris activation during landing. Additionally, sex-based differences in muscle activation were examined, as well as the consistency of activation patterns across the sequence of 5 CMJ, to determine whether it may serve as a test for assessing muscle fatigue.

## 2. Materials and Methods

### 2.1. Participants

A cross-sectional descriptive study was conducted on a sample of 31 subjects (20 women and 11 men, 22.52 ± 3.295 years, BMI 24.32, weight 58.23 ± 4..32). The inclusion criterion was an age between 18 and 30 years. Participation in the study was restricted for subjects with systemic diseases, those who had undergone surgery or had suffered any lower limb injury in the past 12 months, individuals with mental health disorders, those experiencing non-specific pain in the last month, and pregnant women.

The study protocol was approved by the University’s Ethical Research Committee. All subjects received an information sheet and signed an informed consent form. The principles of the 1964 Declaration of Helsinki were fully adhered to throughout the study. Data on sex, age, and laterality were collected. Laterality was determined using three tests from the Harris Test of Lateral Dominance [35]: asking the subject to kick a ball following a pass, to strike a ball with the foot under a chair, and to identify which leg supports first after an unexpected imbalance.

### 2.2. Measures

Surface electromyographic activity was recorded using a FreeEMG wireless system (BTS Bioengineering, Garbagnate Milanese, Italy), consisting of six wireless probes. Electrodes were then placed according to the Seniam protocol [36] after shaving and cleaning the skin with alcohol. Additionally, a Lycra mesh garment was used to secure the electrodes, and it was verified on the signal acquisition screen that this did not introduce any additional noise into the recordings.

For electromyography recordings, a high sampling frequency (1000 Hz) is required to accurately capture the signal.

### 2.3. Procedures

The subjects underwent a 5 min warm-up on a stationary bike at an intensity allowing for easy conversation. Following the warm-up, familiarization with the tests to be performed was conducted, allowing the subject to complete 3 CMJ trials and 1 set of 5 CMJ.

Data were collected from 3 CMJ repetitions, each separated by 30 s. After 5 min of recovery, 3 sequences of 5 CMJ were performed, with a 1 min rest between each sequence.

To precisely determine the phase of the jump at which data were recorded, each trial was captured using BTL. EMG Analyzer [software] V3.1 synchronized with a high-speed camera. Two temporal events were defined for data extraction: take-off and landing. The take-off was identified as the video frame in which the subject lost visual contact with the ground, and landing was defined as the frame in which ground contact was re-established. Data were subsequently extracted from the corresponding time points on the EMG signal, as marked by these predefined events.

The raw signal from the jumps was processed using high-frequency (20 Hz) and low-frequency filtering (400 Hz). The signal was then rectified and smoothed using the Root Mean Square (RMS) method. Activation percentages were obtained using the dynamic normalization method [37]. The average peak activation values in millivolts (mV) from three repetitions of both CMJ and 5 CMJ during the take-off phase and landing phase for each muscle (VM, VL, and BF) were recorded. This value was established as 100% activation for each muscle. Using the activation data for the take-off and landing phases, the percentage of activation for each muscle was determined.

For the calculation of the H/Q ratios, the BF activation value in mV was divided by the values for VM and VL, yielding both ratios.

### 2.4. Statistical Analysis

Data analysis was performed using IBM SPSS Statistics version 28.0.1.1. Descriptive values of the muscle activation and ratios H/Q (H/VM and H/VL) for CMJ and 5 CMJ jumps, both during the take-off and landing phases, are presented as mean, standard deviation, and confidence interval (CI) for variables that followed a normal distribution, and median, range, and confidence interval (CI) when normality was not met. To assess the normality of the electromyographic activation data, the Shapiro–Wilk test was applied. A paired *t*-test was used for variables that followed a normal distribution to determine whether there were differences in muscle activation between the take-off and landing phases of each jump. For the analysis of the ratios, as well as for the BF value during the landing phase of the 5 CMJ (normality was not met), the Wilcoxon test was employed. Furthermore, differences in muscle activation and ratios between the take-off phase of both jumps and the landing phase of both jumps (CMJ and the last jump of 5 CMJ) were examined using an independent *t*-test for normally distributed variables, and the Mann–Whitney test for those that did not exhibit a normal distribution.

Additionally, an independent samples *t*-test was conducted to determine whether there were differences in muscle activation between sexes.

Finally, a repeated measures analysis of variance (ANOVA) was conducted to examine differences in muscle activation across the five jumps of the 5 CMJ protocol. The within-subjects factor was Jump (five levels), and the between-subjects factor was Sex (male vs. female). Prior to interpreting the within-subjects effects, Mauchly’s test of sphericity was performed to assess the assumption of sphericity. In cases where this assumption was violated (*p* < 0.05), the degrees of freedom were corrected using the Greenhouse–Geisser or Huynh–Feldt adjustments, as appropriate. Effect sizes were reported using partial eta squared (η^2^_p_). Post hoc tests and trend analyses (linear, quadratic, and cubic) were applied to explore the nature of any significant effects or interactions.

For all comparisons, the *p*-value (with the significance level set at *p* < 0.05), the 95% confidence interval, and the effect size were calculated: Cohen’s d for parametric tests and r for non-parametric tests.

## 3. Results

Table 1 shows mean and median values of the activation percentages for BF, VM, and VL during the take-off and landing phases of the CMJ and 5 CMJ jumps. It also presents the Shapiro–Wilk test values for each muscle, indicating normative values for the activation percentage of all muscles during both the take-off and landing phases of both jumps, except for BF activation during the landing phase of the 5 CMJ and the activation ratios.

Table 2 shows the results of the paired *t*-test and Wilcoxon test, indicating similar activation of BF during both take-off and landing phases in the CMJ, but different activation in the last jump of the 5 CMJ (Table 2), with t30 = 0.228, *p* = 0.821, d = 0.106 and t30 = 1.930, *p* = 0.054, d = 0.28, respectively. The Wilcoxon test was applied to this variable due to the non-normal.

Figure 1 shows mean muscle activation (±SD) of the vastus medialis (VM), vastus lateralis (VL), and biceps femoris (BF) during the take-off and landing phases of the CMJ. Increased activation of VM and VL was observed during take-off, whereas BF activation remained relatively stable across both phases. The BF/VM and BF/VL activation ratios during the take-off and landing phases of the CMJ are shown in Figure 2, with higher BF/VM ratios observed during the landing phase. These findings may reflect enhanced co-contraction demands for joint stability during impact.

For the VM, VL, and BF/VM ratio, there are statistically significant differences in activation between the take-off and landing phases in both the CMJ and 5 CMJ (last jump). For the BF/VL ratio, significant differences were found in the CMJ, but not in the 5 CMJ (last jump). (Table 2).

The results of the independent *t*-test (Table 3) show slightly higher muscle activation in the take-off and landing phases for the BF, VM, and VL muscles in the final jump of the 5 CMJ compared to the CMJ, but these differences were not statistically significant. Regarding the ratios, they appear to decrease in the final jump of the 5 CMJ compared to the CMJ during the landing phase.

An independent samples *t*-test (Table 4) was conducted to examine differences in muscle activation during the take-off phase of the jump between males and females. The results revealed a statistically significant difference in VM activation in CMJ, t(29) = 3.603, *p* = 0.001. Males demonstrated higher VM activation (M = 0.758, SD = 0.169) compared to females (M = 0.493, SD = 0.208). The 95% confidence interval for the mean difference ranged from 0.114 to 0.415. No significant differences were found during the landing phase; however, the effect size was moderate (d = 0.591).

On the other hand, regarding the final jump of the 5 CMJ, there is a clear difference in VM muscle activation during the take-off phase between males and females, with higher activation observed in males (M = 0.752, SD = 0.183) compared to females (M = 0.524, SD = 0.217), the effect size was height (d = 1.103).

Figure 3 shows males exhibited greater VM activation while females showed greater variability in BF activation. These results suggest potential differences in neuromuscular recruitment strategies between sexes.

No differences were found in the muscle activation patterns across the five jumps of the 5 CMJ in any muscle group, either during the take-off or landing phases (Table 5). Moreover, the repeated measures ANOVA revealed no significant Jump × Sex interaction in any muscle group, except for the activation pattern of the vastus lateralis (VL) during the landing phase, where a significant interaction was observed between males and females, F(4, 25) = 3.464, *p* = 0.010. Specifically, activation was less variable among females, whereas in males, a noticeable reduction in VL activation was evident in the final jump.

## 4. Discussion

The results of this study show a normal distribution in the muscle activation of BF, VM and VL during the take-off and landing phases of the CMJ and final jump in a 5 CMJ sequence. This is one of the few studies that specifically measures muscle activation during these particular phases of the jump and is in line with the research by Cavanaugh et al. (2017), which reported very high reliability in vertical jump measures performed with a single leg [8]. Additionally, Ebben et al. (2010) [16] and Ellenberger et al. (2021) [18], showed results similar to ours, despite using a different method for normalizing the EMG signal. Both authors employed a normalization method based on expressing muscle activation at each event or time point as a percentage of the peak EMG activity recorded during the entire task. This approach replaces the traditional method of reporting EMG values relative to a maximal voluntary isometric contraction (MVIC) and is typically adopted in situations where it is challenging to obtain reliable and standardized maximal contractions. In addition, our study presents more precise results than those reported by Ebben et al. (2010) for BF, since we present narrower CI to 95% and lower SD [16]. These findings make it the fastest and most accurate method to obtain and interpret the muscle activation pattern in CMJ and the final jump of 5 CMJ.

On the other hand, Goodwin et al. (1999) did not obtain reliable results for BF activation during the take-off phase [38]. This aligns with the findings of this study, where normal activation was observed in the BF during the take-off phase, but not during the landing phase, where normality was only observed in the CMJ. Our findings complement current evidence by establishing normal CMJ values during the take-off phase. However, we have not been able to establish normality during the landing phase in the BF, so further research will be necessary to determine the cause of this event.

Our results show greater activation of the biceps femoris (BF) during the landing phase in females compared to males, which contrasts with the findings of Ebben et al., who demonstrated a lower BF activation in the landing phase of jumps in women compared to men [16]. However, Ellenberger et al. (2021) found that male skiers tended to show lower hamstring activation values [18]. This latter author is more in line with our findings, as our study revealed sex-related differences in muscle activation. Specifically, greater activation of the vastus medialis (VM) was observed in males during the take-off phase of both the CMJ and 5 CMJ jumps. However, females exhibited a higher biceps femoris to vastus medialis (BF/VM) ratio during the landing phase. This finding could be justified by the architecture and muscle activation difference between sexes, as they play an important role in determining the BF/VM and BF/VL ratios [39]

This may be explained by the findings of Padulo et al. (2013), who determined that there is a reduction in BF muscle activation due to the biphasic coupling of concentric and eccentric contractions during the stretch–shortening cycle, as the energy stored in elastic tissues during the stretch phase is released during the shortening phase, thus reducing the need for greater muscle activation in the latter [40].

Regarding the BF/VM and BF/VL ratios, these did not follow a normal distribution in any of the jump phases. This might be related to the fact that some individuals are more prone than others to knee injuries, with it being well-established that certain values of these ratios can indicate a higher or lower predisposition to such injuries [14,41].

Regarding the differences observed between take-off and landing phases, we found that the activation of the Vasti muscles is much greater during the take-off phase, which aligns with the findings of Cerrah et al. (2014), who also reported maximum activation for these structures in the middle of the mentioned phase [42]. In contrast, the activation of the BF remained almost unchanged during this phase in our sample, which is consistent with the findings of Cone et al. (2021), who confirmed this difference between phases by noting a greater asymmetry in the distribution of vertical ground reaction force during the landing phase compared to the take-off phase [43]. In their study, Baratta et al. (1988) also observed greater activity of the quadriceps group, including the rectus femoris (RF), than the hamstring group during a functional movement, confirming the activation data from both studies indicating some co-activation of these muscle groups to provide necessary knee joint stability [44].

As for the differences between jumps (CMJ and 5 CMJ), Behm et al. (2010) demonstrated how fatigue can affect activation, particularly the intensity of the signal and co-activation, with a decrease in intensity and intermuscular coordination [45]. In our case, although activation was slightly higher in the last jump of 5 CMJ, the differences with CMJ were not statistically significant. It is possible that performing 5 consecutive CMJs is not sufficient to induce appreciable muscle fatigue, and a change in muscle activation might have been observed if 10 CMJs had been performed. Furthermore, the analysis of the muscle activation pattern across the five jumps of the 5 CMJ revealed a generally consistent activation profile, with no substantial changes observed, except in the vastus lateralis (VL) during the landing phase. Regarding the influence of sex on this pattern, only the activation of the VL and vastus medialis (VM) during landing showed noticeable differences. Specifically, a decline in VM activation was observed in males during the final jump, whereas females maintained a more stable activation profile throughout. In the case of the VL during landing, the activation pattern in males appeared irregular, with alternating increases and decreases across successive jumps.

Bermúdez and Fábrica (2014) analyzed muscle activation during a 5 CMJ under fatigue conditions, measuring it based on 1 min of continuous jumping quantified by the drop in average power analyzed in 15 s intervals, revealing advances in peak VM activation along with changes in RF and BF in the electromyographic analysis associated with a force platform [46].

Our study did not consider the biomechanical variables of the subjects, which is a limitation, as variability in the kinematics of the hip, knee, and ankle joints may cause changes in muscle activation. The biomechanics of these three joints present different kinematic patterns between the sexes, as observed by Earl et al. (2007), where women showed greater hip internal rotation, which could alter muscle activation in the lower limbs [47]. Additionally, Russell et al. (2006) observed that women tend to land jumps with greater knee valgus, which may also affect muscle activity [48]. Furthermore, factors such as hamstring stiffness, which we also did not assess in this study, could lead to increased activation of these muscles during the landing phase of the vertical jump, as suggested by Jankaew et al. (2023) [49]. On the other hand, establishing normative values of muscle activation during the execution of a well-studied jump used for injury assessment, such as the countermovement jump (CMJ) [14,50,51], may serve as a reference point for determining injury risk in cases of either increased or decreased muscular activation. For instance, reduced hamstring activation in comparison to quadriceps activation could indicate a higher risk of anterior cruciate ligament (ACL) injury in the knee.

In patients undergoing rehabilitation, normative values can provide insight into the progression of neuromuscular recovery and assist in decision-making regarding return-to-play timing. In the case of the 5 CMJ protocol, deviations from normative activation patterns may help monitor neuromuscular status and fatigue levels in conjunction with other variables [50,52].

This objective assessment of fatigue, inferred from variations in muscle activation relative to normative values, could represent a starting point for load management planning based on the type of fatigue detected [51].

## 5. Conclusions

The activation values of the anterior and posterior thigh muscle groups follow normal patterns in the population studied (individuals aged 18 to 30 years), with greater activation observed in the take-off phase for the vastus medialis (VM) and vastus lateralis (VL) compared to the landing phase. In contrast, the biceps femoris (BF) exhibited similar activation levels in both the take-off and landing phases.

Males and females exhibit differences in vastus medialis (VM) activation during the landing phase of the jumps, with females activating the VM and vastus lateralis (VL) to a similar extent, whereas males show greater activation of the VM compared to the VL.

The 5 CMJ protocol does not appear to induce greater muscle fatigue than the CMJ in untrained individuals aged 18 to 30 years, in both males and females.

Further research is needed to determine normal activation values for the jump, considering sex differences, joint range of motion of the lower limbs, and hamstring extensibility.

## Figures and Tables

**Figure 1 bioengineering-12-00767-f001:**
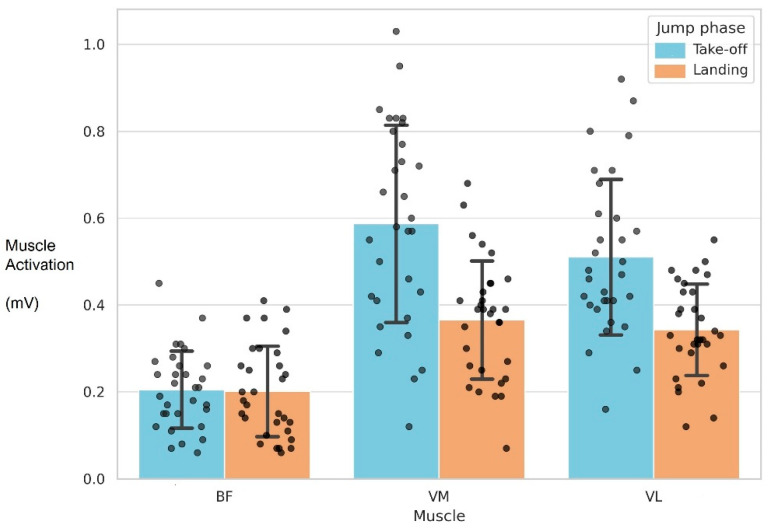
Average activation of the BF, VM, and VL muscles during the take-off and landing phases of the CMJ, with error bars representing the standard deviation.

**Figure 2 bioengineering-12-00767-f002:**
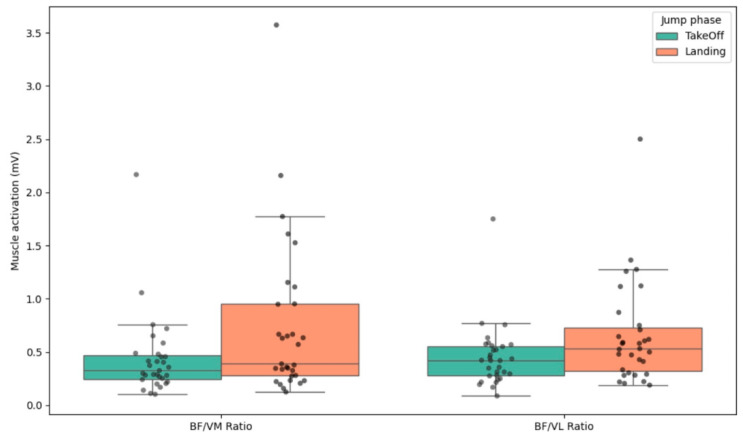
BF/VM and BF/VL activation ratios during the take-off and landing phases of the CMJ.

**Figure 3 bioengineering-12-00767-f003:**
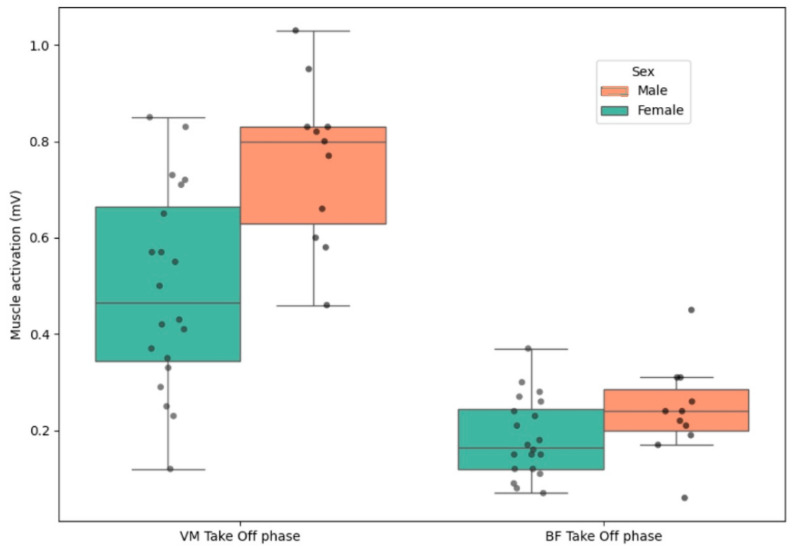
Sex-based differences in VM and BF activation during the take-off phase of the CMJ.

**Table 1 bioengineering-12-00767-t001:** Muscle activation in the Countermovement Jump (CMJ) and 5 consecutive CMJs (5 CMJ) Mean (M), standard deviation (SD) and confidence interval values for Mean (M), Standard Deviation (SD), Biceps Femoris (BF); Vastus Medialis (VM); Vastus Lateralis (VL); Biceps Femoris/Vastus Medialis Rate (R BF/VM); Biceps Femoris/Vastus Lateralis Rate (R BF/VL).

		Take-Off				Landing			
		M ± SD	95% IC	W	*p*	M ± SD	95% IC	W	*p*
BF	CMJ	0.20 ± 0.09	0.172–0.239	0.970	0.532	0.20 ± 0.10	0.162–0.24	0.935	0.058
	5 CMJ	0.21 ± 0.10	0.179–0.254	0.954	0.115	0.18 ± 0.11	0.141–0.223	0.892	0.005
VM	CMJ	0.58 ± 0.23	0.502–0.672	0.978	0.752	0.36 ± 0.14	0.315–0.417	0.974	0.645
	5 CMJ	0.60 ± 0.23	0.52–0.69	0.952	0.181	0.38 ± 0.13	0.332–0.431	0.971	0.542
VL	CMJ	0.51 ± 0.18	0.443–0.576	0.961	0.31	0.34 ± 0.10	0.303–0.381	0.977	0.727
	5 CMJ	0.53 ± 0.19	0.461–0.606	0.938	0.73	0.37 ± 0.12	0.327–0.419	0.937	0.069
R BF/VM	CMJ	0.43 ± 0.39	0.292–0.581	0.634	<0.001	0.74 ± 0.72	0.473–1.00	0.752	0.001
	5 CMJ	0.41 ± 0.26	0.336–0.597	0.856	<0.001	0.58 ± 0.48	0.410–0.767	0.829	0.001
R BF/VL	CMJ	0.45 ± 0.30	0.342–0.566	0.709	<0.001	0.65 ± 0.46	0.486–0.822	0.801	0.001
	5 CMJ	0.46 ± 0.35	0.336–0.597	0.668	<0.001	0.54 ± 0.47	0.369–0.720	0.665	0.001

**Table 2 bioengineering-12-00767-t002:** Muscle activation during the take-off and landing phases: Mean (M), Standard Deviation (SD), median (Mdn), Biceps Femoris (BF), Vastus Medialis (VM), Vastus Lateralis (VL).

		Take Off	Landing Jump	95% Confidence Interval of the Difference				
		**M ± SD**	**M ± SD**	**Lower**	**Upper**	**T**	** *P* **	**d**
BF	CMJ	0.20 ± 0.09	0.20 ± 0.10	−0.035	0.043	0.228	0.821	0.106
	5 CMJ	0.21 ± 0.10	0.18 ± 0.11	−0.003	0.073	1.930	0.054	0.28
VM	CMJ	0.58 ± 0.23	0.36 ± 0.14	0.169	0.273	8.667	0.000	0.142
	5 CMJ	0.60 ± 0.23	0.38 ± 0.13	0.170	0.277	8.501	0.000	0.146
VL	CMJ	0.51 ± 0.18	0.34 ± 0.10	0.219	0.219	6.620	0.000	0.14
	5 CMJ	0.53 ± 0.19	0.37 ± 0.12	0.108	0.213	6.248	0.000	0.143
		**Mdn (Range)**	**Mdn (Range)**			**Z**	** *P* **	** *r* **
R BF/VM	CMJ	0.33 (2.13)	0.39 (3.30)	0.383	0.816	3.63	0.001	0.651
	5 CMJ	0.33 (1.30)	0.35 (1.65)	0.146	0.709	2.646	0.004	0.475
R BF/VL	CMJ	0.40 (1.74)	0.53 (2.19)	0.146	0.709	2.65	0.008	0.475
	5 CMJ	0.41 (1.97)	0.44 (2.59)	−0.193	0.496	0.958	0.445	0.172

**Table 3 bioengineering-12-00767-t003:** Muscle activation at take-off phases of both jumps and the landing phases measured using independent t-tests and Mann–Whitney U test. Mean (M), Standard Deviation (SD), median (Mdn), Biceps Femoris (BF), Vastus Medialis (VM), Vastus Lateralis (VL).

		**CMJ**	**5 CMJ**				**95% Confidence Interval** ** of the Difference**	

		**M ± SD**	**M ± SD**	**t**	** *p* **	**d**	**Lower**	**Upper**
BF	Take off	0.20 ± 0.09	0.21 ± 0.10	−0.457	0.707	0.096	−0.060	0.038
	Landing	0.20 ± 0.10	0.18 ± 0.11	0.708	0.954	0.11	−0.035	0.075
VM	Take off	0.58 ± 0.23	0.60 ± 0.23	−0.312	0.817	0.23	−0.136	0.099
	Landing	0.36 ± 0.14	0.38 ± 0.13	−0.454	0.961	0.13	−0.085	0.053
VL	Take off	0.51 ± 0.18	0.53 ± 0.19	−0.499	0.742	0.19	−0.121	0.073
	Landing	0.34 ± 0.10	0.37 ± 0.12	−1.041	0.416	0.34	−0.090	0.028
		**Mdn (Range)**	**Mdn (Range)**	**U**	** *p* **	** *r* **	**Lower**	**Upper**
R BF/VM	Take off	0.43 ± 0.39	0.41 ± 0.26	495.00	0.838	0.026	−0.214	0.274
	Landing	0.74 ± 0.72	0.58 ± 0.48	415.00	0.356	−0.117	−0.360	0.135
R BF/VL	Take off	0.45 ± 0.30	0.46 ± 0.35	473.00	0.916	−0.11	−0.262	0.237
	Landing	0.65 ± 0.46	0.54 ± 0.47	382.50	0.168	−0.175	−0.407	0.078

**Table 4 bioengineering-12-00767-t004:** Differences in muscle activation during the take-off phase of the jump between males and females: Mean (M), Standard Deviation (SD), Biceps Femoris (BF), Vastus Medialis (VM), Vastus Lateralis (VL).

		CMJ				
		**Male**	**Female**			
		**M ± SD**	**M ± SD**	**t**	** *P* **	**D**
BF	Take off	0.243 ± 0.097	0.186 ± 0.081	1.743	0.092	0.659
	Landing	0.155 ± 0.089	0.227 ± 0.107	−1.870	0.072	−0.709
VM	Take off	0.758 ± 1.690	0.493 ± 0.208	3.603	0.001	0.272
	Landing	0.418 ± 0.091	0.337 ± 0.154	1.838	0.077	0.591
VL	Take off	0.552 ± 0.221	0.486 ± 0.157	0.974	0.338	0.367
	Landing	0.358 ± 0.132	0.334 ± 0.091	0.614	0.544	0.227
		**5 CMJ (last jump)**				
		**Male**	**Female**	**Jump**		
		**M ± SD**	**M ± SD**	**t**	** *P* **	**D**
BF	Take off	0.257 ± 0.085	0.195 ± 0.105	1.685	0.103	0.626
	Landing	0.153 ± 0.105	0.198 ± 0.114	−1.089	0.285	0.405
VM	Take off	0.752 ± 0.183	0.524 ± 0.217	2.950	0.006	1.103
	Landing	0.417 ± 0.777	0.362 ± 0.155	1.324	0.196	0.120
VL	Take off	0.593 ± 0.226	0.501 ± 0.179	1.242	0.224	0.471
	Landing	0.381 ± 0.137	0.369 ± 0.123	0.258	0.798	0.094

**Table 5 bioengineering-12-00767-t005:** Repeated measures ANOVA results for the main effect of Jump and the Jump × Sex interaction for the vastus medialis (VM), vastus lateralis (VL), and biceps femoris (BF) during the take-off and landing phases of the 5th jump of 5 CMJ. Greenhouse–Geisser corrections were applied where the assumption of sphericity was violated. F and *p*-values are reported along with partial eta squared (η^2^_p_) as a measure of effect size.

		1º Jump		2º Jump		3º Jump		4º Jump		5º Jump		Multivariant Test			Multivariant Test		
		Male	Female	Male	Female	Male	Female	Male	Female	Male	Female	Jump			Sex		
		M ± SD	M ± SD	M ± SD	M ± SD	M ± SD	M ± SD	M ± SD	M ± SD	M ± SD	M ± SD	F	*p*	ŋ^2^	F	*p*	ŋ^2^
BF	Take off	0.221 ± 0.01	0.670 ± 0.10	0.318 ± 0.127	0.808 ± 0.858	0.304 ± 0.126	1.049 ± 1.241	0.311 ± 0.111	1.010 ± 1.261	0.300 ± 0.121	0.627 ± 0.749	0.890	0.472	0.031	0.329	0.858	0.012
	Landing	0.148 ± 0.080	0.245 ± 0.160	0.169 ± 0.116	0.219 ± 0.122	0.150 ± 0.108	0.258 ± 0.157	0.177 ± 0.082	0.271 ± 0.171	0.160 ± 0.119	0.254 ± 0.168	0.999	0.411	0.034	0.690	0.600	0.034
VM	Take off	0.652 ± 0.169	0.447 ± 0.258	0.695 ± 0.184	0.453 ± 0.231	0.627 ± 0.313	0.481 ± 0.254	0.695 ± 0.225	0.466 ± 0.246	0.701 ± 0.230	0.496 ± 0.270	1.577	0.185	0.053	0.333	0.855	0.012
	Landing	0.437 ± 0.162	0.322 ± 0.149	0.465 ± 0.180	0.334 ± 0.150	0.469 ± 0.168	0.318 ± 0.140	0.456 ± 0.189	0.326 ± 0.140	0.371 ± 0.126	0.342 ± 0.151	1.757	0.143	0.059	3.464	0.010	0.110
VL	Take off	0.589 ± 0.185	0.436 ± 0.211	0.582 ± 0.255	0.457 ± 0.234	0.638 ± 0.235	0.482 ± 0.222	0.582 ± 0.220	0.466 ± 0.234	0.626 ± 0.215	0.463 ± 0.233	1.581	0.184	0.054	0.445	0.776	0.016
	Landing	0.406 ± 0.191	0.316 ± 0.151	0.508 ± 0.181	0.364 ± 0.182	0.395 ± 0.137	0.378 ± 0.199	0.444 ± 0.186	0.361 ± 0.166	0.394 ± 0.125	0.373 ± 0.176	3.521	0.010	0.112	3.250	0.015	0.104

## Data Availability

Data are available upon request from the corresponding author.

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
