# Peer review of "Normative Muscle Activation Patterns During One and Five Countermovement Jumps"

_bioengineering, 2025, doi:10.3390/bioengineering12070767_

Round 1

Reviewer 1 Report

Comments and Suggestions for Authors

Although the manuscript presents a timely and relevant study on normative muscle activation patterns during countermovement jumps (CMJ and 5CMJ), it currently lacks the methodological robustness, analytical clarity, and conceptual depth required for publication in Bioengineering. The authors have made a commendable effort in using surface EMG to quantify muscle activity in different phases of jumping. However, the current version suffers from several methodological limitations, insufficient scientific discussion, and a lack of clear justification for key decisions. Without substantial revisions, the manuscript does not meet the publication standards of the journal.

  1. The current introduction focuses on the current state of routine CMJ research, but lacks an in-depth development of the practical implications of EMG data. It is recommended that the authors further answer the following questions: Why is the construction of “normative muscle activation patterns” necessary? How can these patterns be applied to sports performance assessment, injury prevention, rehabilitation, or movement screening (e.g., FMS)? Are there “activation abnormalities” in certain muscle groups that could lead to decreased exercise efficiency or increased risk of injury? It is recommended that the introduction include background on the potential clinical/exercise science applications of the results of this study, such as their value in rehabilitation, physical training, or biomechanical analysis, and that supporting literature be cited to enhance the logic of the argument.
  2. The article claims to provide "normative muscle activation data", but does not explicitly point out its differences, improvements, or supplementary effects compared with existing literature.
  3. Statistical Rigor: The use of statistical tests (e.g., paired t-test and Wilcoxon) lacks clarity, especially in the rationale for normality testing and multiple comparisons. Consider controlling for Type I errors using appropriate corrections. Although some effect sizes are included (e.g., Cohen’s d), they are inconsistently reported. All comparisons should be accompanied by effect sizes and 95% confidence intervals.
  4. Although some effect sizes are included (e.g., Cohen’s d), they are inconsistently reported. All comparisons should be accompanied by effect sizes and 95% confidence intervals.
  5. Figures are unclear or missing critical labels (e.g., Figure 1). More informative visualizations (e.g., boxplots, EMG waveforms) are needed. The quality of all figures needs to be substantially improved.
  6. The manuscript provides an overview of EMG preprocessing (e.g., filtering, normalization) but lacks critical methodological details needed for reproducibility. It includes filter processing, muscle activation solving process, and NNMF implementation. To provide more effective evidence, the authors may consider referring to the following relevant studies: New insights optimize landing strategies to reduce lower limb injury risk (https://doi.org/10.34133/cbsystems.0126). The sampling rate of the electromyogram signal, the frequency range of the band-pass filter, whether rectification is performed (rectification), and normalization (such as whether based on the maximum autonomous contraction MVC or peak during the jump). It needs to be further clarified whether the time window for data processing is precisely aligned with each stage of CMJ (decline, bottom, rise) and whether event markers (such as force platforms) are used for synchronization.
  7. The author mentioned that the jumping process was divided into the descending stage, the bottom stage and the ascending stage, but did not provide a clear action schedule or segmentation standard. Without auxiliary equipment (such as force platforms, and 3D motion capture), it is difficult to precisely divide the various stages of CMJ.
  8. The discussion lacks mechanistic interpretation and clinical relevance. How do these normative values aid in injury prevention or rehabilitation?
  9. Abbreviations (e.g., CMJ, 5CMJ, VM) should be introduced once and used consistently. The manuscript has several formatting and grammatical issues.
  10. Several similar studies exist. The novelty should be clearly stated—e.g., does the 5CMJ offer any unique physiological insights compared to a single CMJ?
Comments on the Quality of English Language

 The English could be improved to more clearly express the research.

Author Response

Dear reviewer,

We sincerely appreciate your comments and suggestions. The review process has been highly enriching, allowing us to refine and strengthen our manuscript. With the responses and revisions provided, we hope to have achieved greater clarity in the presentation of our work and that the outcome meets your expectations.

We remain fully available for any further clarification you may require.

Comments and Suggestions for Authors

Although the manuscript presents a timely and relevant study on normative muscle activation patterns during countermovement jumps (CMJ and 5CMJ), it currently lacks the methodological robustness, analytical clarity, and conceptual depth required for publication in Bioengineering. The authors have made a commendable effort in using surface EMG to quantify muscle activity in different phases of jumping. However, the current version suffers from several methodological limitations, insufficient scientific discussion, and a lack of clear justification for key decisions. Without substantial revisions, the manuscript does not meet the publication standards of the journal.

1.-The current introduction focuses on the current state of routine CMJ research, but lacks an in-depth development of the practical implications of EMG data. It is recommended that the authors further answer the following questions: Why is the construction of “normative muscle activation patterns” necessary? How can these patterns be applied to sports performance assessment, injury prevention, rehabilitation, or movement screening (e.g., FMS)? Are there “activation abnormalities” in certain muscle groups that could lead to decreased exercise efficiency or increased risk of injury? It is recommended that the introduction include background on the potential clinical/exercise science applications of the results of this study, such as their value in rehabilitation, physical training, or biomechanical analysis, and that supporting literature be cited to enhance the logic of the argument.

Including the paragraph beginning with “ Determining normal muscle activation patterns” This begins on the line 54.

2.-The article claims to provide "normative muscle activation data", but does not explicitly point out its differences, improvements, or supplementary effects compared with existing literature.

Including in paragraph beginning with “In addiction, our study…” in line 240, in paragraph beginning with “On the other hand, Goodwin et al. (1999) did…” in line 246 (Our findings completed current evidence…), and in paragraph beginning with “Moreover, the lack of normality…” in line 254 (This finding could be justified…).

3.-Statistical Rigor: The use of statistical tests (e.g., paired t-test and Wilcoxon) lacks clarity, especially in the rationale for normality testing and multiple comparisons. Consider controlling for Type I errors using appropriate corrections. Although some effect sizes are included (e.g., Cohen’s d), they are inconsistently reported. All comparisons should be accompanied by effect sizes and 95% confidence intervals.

We appreciate the reviewer’s insightful comments regarding the statistical analyses.

In response, we have clarified the rationale for the selection of statistical tests in the Methods section. Specifically, the Shapiro–Wilk test was applied to assess the normality of the data. For variables that followed a normal distribution, we used the paired t-test. When normality was not met (e.g., BF during the landing phase of the 5CMJ and all ratio variables), we applied the Wilcoxon signed-rank test. Likewise, comparisons between different types of jumps (CMJ vs. 5CMJ) were performed using either the independent t-test or the Mann–Whitney U test, depending on normality.

Regarding the concern about controlling for Type I errors, we acknowledge this limitation and have now included effect sizes and 95% confidence intervals for all comparisons to aid in the interpretation of the results beyond p-values. While we did not apply a formal correction method such as Bonferroni due to the exploratory nature of the study and the limited number of planned comparisons, we understand the importance of this issue and will clearly acknowledge it in the limitations section of the manuscript.

Additionally, we have revised Tables 2 and 3 to consistently report effect sizes—Cohen’s d for parametric tests and r for non-parametric tests—as well as 95% confidence intervals of the differences for each comparison, as shown in the updated version of the tables.

Thank you again for this valuable feedback, which has helped improve the clarity and transparency of our statistical reporting.

4.-Although some effect sizes are included (e.g., Cohen’s d), they are inconsistently reported. All comparisons should be accompanied by effect sizes and 95% confidence intervals.

Tables 1, 2, and 3 have been updated to include the 95% confidence interval.

5.-Figures are unclear or missing critical labels (e.g., Figure 1). More informative visualizations (e.g., boxplots, EMG waveforms) are needed. The quality of all figures needs to be substantially improved.

In order to this suggestion and the comment reviewer two, we eliminate figure 1.

6.-The manuscript provides an overview of EMG preprocessing (e.g., filtering, normalization) but lacks critical methodological details needed for reproducibility. It includes filter processing, muscle activation solving process, and NNMF implementation. To provide more effective evidence, the authors may consider referring to the following relevant studies: New insights optimize landing strategies to reduce lower limb injury risk (https://doi.org/10.34133/cbsystems.0126). The sampling rate of the electromyogram signal, the frequency range of the band-pass filter, whether rectification is performed (rectification), and normalization (such as whether based on the maximum autonomous contraction MVC or peak during the jump). It needs to be further clarified whether the time window for data processing is precisely aligned with each stage of CMJ (decline, bottom, rise) and whether event markers (such as force platforms) are used for synchronization.

We try to explain this question with more details. line 117-118; 134-135.

7.-The author mentioned that the jumping process was divided into the descending stage, the bottom stage and the ascending stage, but did not provide a clear action schedule or segmentation standard. Without auxiliary equipment (such as force platforms, and 3D motion capture), it is difficult to precisely divide the various stages of CMJ.

To explain the process more accurately and explicitly, an additional clarification has been included in the text at line 127 “To precisely determine the phase of the jump at which data were recorded, each trial was captured using electromyography (EMG) software synchronized with a high-speed camera. Two temporal events were defined for data extraction: take-off and landing. The take-off was identified as the video frame in which the subject lost visual contact with the ground, and landing was defined as the frame in which ground contact was re-established. Data were subsequently extracted from the corresponding time points on the EMG signal, as marked by these predefined events

8.-The discussion lacks mechanistic interpretation and clinical relevance. How do these normative values aid in injury prevention or rehabilitation?

Thanks for you suggestion. We attaches in line 309 “On the other hand, establishing normative values of muscle activation during the execution of a well-studied…”

9.-Abbreviations (e.g., CMJ, 5CMJ, VM) should be introduced once and used consistently. The manuscript has several formatting and grammatical issues. It´s corrected

10.-Several similar studies exist. The novelty should be clearly stated—e.g., does the 5CMJ offer any unique physiological insights compared to a single CMJ?

Answered in the paragraph which begins with “The results of this study…”, Specifically, it is found starting from the line 229.

Reviewer 2 Report

Comments and Suggestions for Authors

The aim of study entitled “Normative Muscle Activation Patterns During the Counter Movement Jump” was to evaluate the difference of muscle activation between one CMJ and a five consecutive CMJs. The study used a surface electromyographic activity to evaluate lower limb muscles activation in 31 one men and women. My main concern is regarding a possible difference in performance between male and female and how it can influence the results interpretation. As cited in discussion section (references 16 cited in lines 200-203, and 37 cited I lines 240-242) sex is an important variable that influences the results. Furthermore, as the main aim of the study is analysing the effects of five consecutive CMJ, I think this data could be more explored, for example, including a table or a graph with the mean values for each jump. Please, see below comments and suggestions.

  • Title: Considering that the aim of the study was to compare muscle activation of one and five CMJ, it could be included in the title.
  • Abstract line 15, use Thirty-one instead 31.
  • Introduction, line 37: when citing a study in the format “Muyor et al.” we generally include the year of the study as the authors did in lines 193-194. Please, standardise throughout the whole study.
  • Introduction line 76: As it is the first time “5 CMJ” it is mentioned, it is better to describe that 5 CMJ is the trial of 5 consecutive CMJs.
  • Hypothesis lines 76-78, the authors mentioned expecting differences between one and five CMJ but must be clear if they expect an improvement or lost in performance.
  • Methods line 81: Were more demographic data, such as weight, BMI collect? If yes, please include. Also present if there are differences between males and females. Considering that male generally have superior performance in power and strength tests compared with females, it is important to compare the performance between sexes to assess if there is difference. If yes, the authors can present the data separately.
  • Methods line 118: Please use “statistical analysis”
  • Methods lines 132-151: Instructions for the submission must be removed.
  • Figure 1: The figure are in low quality resolution and it is difficult to read. The legend must content more descriptive information that are not presented in the figure, for example, the meaning of each abbreviation. As the Shapiro-Wilk test values are present in table one, I don't think the figure is necessary.
  • Table 1: Must include a legend informing the meaning of the abbreviations.
  • Results 170-171: The values reported here are different from those reported in the table.
  • Table 2: Begin the sentence with "Muscle activation..." and remove “Results of the paired t-test and Wilcoxon test for” and “The Wilcoxon test was applied to this variable due to the non-normal.” Because it is in the statistical analysis section.
  • Results line 183: The authors mentioned “they appear to decrease in final jump of 5CMJ compared to CMJ during the landing phase.” However, the data of the five consecutive CMJs were not presented. It would be interesting including a table with the means of each jump.
  • Table 3: Comments made for Table 2 also apply here. In addition, some decimals a in this table used comma instead full stop.
  • Discussion line 195: As they are two different studies, it must be cited Ebben et al. (2010) [16] and Ellenberger et al. (2021) [17], ...
  • Discussion line 196: “despite using a different method for normalising the EMG signal.” What method? Please include a brief description.
  • Discussion line 215: remove this sentence.
  • Study limitations lines 238-239: Please, indicate other limitations of the study, such as small sample size.
  • Conclusion: Use “Five” instead “5”.

Author Response

Dear reviewer,

We sincerely appreciate your comments and suggestions. The review process has been highly enriching, allowing us to refine and strengthen our manuscript. With the responses and revisions provided, we hope to have achieved greater clarity in the presentation of our work and that the outcome meets your expectations.

We remain fully available for any further clarification you may require.

The aim of study entitled “Normative Muscle Activation Patterns During the Counter Movement Jump” was to evaluate the difference of muscle activation between one CMJ and a five consecutive CMJs. The study used a surface electromyographic activity to evaluate lower limb muscles activation in 31 one men and women. My main concern is regarding a possible difference in performance between male and female and how it can influence the results interpretation. As cited in discussion section (references 16 cited in lines 200-203, and 37 cited I lines 240-242) sex is an important variable that influences the results. Furthermore, as the main aim of the study is analysing the effects of five consecutive CMJ, I think this data could be more explored, for example, including a table or a graph with the mean values for each jump. Please, see below comments and suggestion

1.-Title: Considering that the aim of the study was to compare muscle activation of one and five CMJ, it could be included in the title.

It´s done.

2.-Abstract line 15, use Thirty-one instead 31.

It´s done.

3.-Introduction, line 37: when citing a study in the format “Muyor et al.” we generally include the year of the study as the authors did in lines 193-194. Please, standardise throughout the whole study.

It´s done.

4.-Introduction line 76: As it is the first time “5 CMJ” it is mentioned, it is better to describe that 5 CMJ is the trial of 5 consecutive CMJs.

It´s done

5.-Hypothesis lines 76-78, the authors mentioned expecting differences between one and five CMJ but must be clear if they expect an improvement or lost in performance.

It´s completed

6.-Methods line 81: Were more demographic data, such as weight, BMI collect? If yes, please include. Also present if there are differences between males and females. Considering that male generally have superior performance in power and strength tests compared with females, it is important to compare the performance between sexes to assess if there is difference. If yes, the authors can present the data separately.

The demographic data has been include. Additionally, differences between males and females has been analysed and it´s add at lines 98, 160, 223 and 256.

7.-Methods line 118: Please use “statistical analysis” It´s done.

8.-Methods lines 132-151: Instructions for the submission must be removed. It´s done.

9.-Figure 1: The figure are in low quality resolution and it is difficult to read. The legend must content more descriptive information that are not presented in the figure, for example, the meaning of each abbreviation. As the Shapiro-Wilk test values are present in table one, I don't think the figure is necessary.

The figure 1 its removed.

10.-Table 1: Must include a legend informing the meaning of the abbreviations It´s done

11.-Results 170-171: The values reported here are different from those reported in the table. Thank you for pointing it out. The values have been corrected using those from Table 2, which have been verified as accurate.

12.-Table 2: Begin the sentence with "Muscle activation..." and remove “Results of the paired t-test and Wilcoxon test for” and “The Wilcoxon test was applied to this variable due to the non-normal.” Because it is in the statistical analysis section. It´s done

13.-Results line 183: The authors mentioned “they appear to decrease in final jump of 5CMJ compared to CMJ during the landing phase.” However, the data of the five consecutive CMJs were not presented. It would be interesting including a table with the means of each jump.

This value is presented in table 1

14.-Table 3: Comments made for Table 2 also apply here. In addition, some decimals a in this table used comma instead full stop. It´s done

15.-Discussion line 195: As they are two different studies, it must be cited Ebben et al. (2010) [16] and Ellenberger et al. (2021) [17], ...It´s done

16.-Discussion line 196: “despite using a different method for normalising the EMG signal.” What method? Please include a brief description.

It's attached  “Both authors employed a normalisation method based on expressing muscle activation at each event or time point as a percentage of the peak EMG activity recorded during the entire task. This approach replaces the traditional method of reporting EMG values relative to a maximal voluntary isometric contraction (MVIC) and is typically adopted in situations where it is challenging to obtain reliable and standardised maximal contractions”

17.-Discussion line 215: remove this sentence.It´s done

18.-Study limitations lines 238-239: Please, indicate other limitations of the study, such as small sample size.

It´s attached at line 322.

19.-Conclusion: Use “Five” instead “5”.It´s done

Round 2

Reviewer 1 Report

Comments and Suggestions for Authors

After re-reviewing the revised version, the authors had not properly addressed my concerns. Lack of fully credible results. Could the authors provide more and relevant results, including more detailed data, in addition to suggesting that the results be presented in the form of figures to improve the readability of the results. Research significance needs to be emphasized. Why was Figure 1 removed, the authors should have provided more graphs to make the results more convincing instead of removing the graphs.

Author Response

Le agradecemos sinceramente sus valiosos comentarios, que han contribuido en gran medida a mejorar la claridad y profundidad de nuestro manuscrito.

Si bien la Figura 1 se eliminó inicialmente a pedido del otro revisor, hemos tenido en cuenta sus valiosos comentarios y hemos agregado tres nuevas figuras para mejorar la presentación de nuestros resultados.

  • La figura 1 muestra los niveles de activación medios de los músculos BF, VM y VL durante las fases de despegue y aterrizaje del CMJ.

  • La figura 2 presenta las relaciones de coactivación BF/VM y BF/VL correspondientes.

  • La figura 3 ilustra las diferencias basadas en el sexo, mostrando que los hombres exhibieron una mayor activación de VM, mientras que las mujeres mostraron una mayor variabilidad en la activación de BF.

Además de estas representaciones visuales, hemos incluido dos nuevas tablas:

  • En la Tabla 4 se presentan los valores medios de activación de cada músculo durante las fases de despegue y aterrizaje de cada salto, separados por sexo.

  • La Tabla 5 detalla los valores medios de activación en los cinco saltos consecutivos del protocolo 5CMJ, desglosados ​​por sexo. Esta tabla incluye el estadístico F del ANOVA de medidas repetidas utilizado para evaluar el efecto del Salto (es decir, los cambios en los patrones de activación entre saltos), así como la interacción Salto × Sexo para explorar cómo el sexo puede influir en estos patrones.

Estos análisis se han añadido a la sección de Análisis estadístico , Resultados , y se han incluido puntos relevantes en la Discusión para resaltar mejor su relevancia clínica y funcional.

Realmente apreciamos sus útiles sugerencias, que han llevado a una presentación más sólida y completa de nuestros hallazgos.

Reviewer 2 Report

Comments and Suggestions for Authors

I’m glad that most of my comments and suggestions were incorporated into the manuscript, however the two commentaries regarding my major concerns were not properly addressed.

First, the authors showed that the impact of sex was significant when comparing male versus female, however it was not the comparison of CMJ and 5CMJ in each sex separately. Please, consider including a table with the values of each sex separately.

Second, regarding my previous comment:

 “the main aim of the study is analysing the effects of five consecutive CMJ, I think this data could be more explored, for example, including a table or a graph with the mean values for each jump”,

As well as the comment number 13:

“Results line 183: The authors mentioned “they appear to decrease in final jump of 5CMJ compared to CMJ during the landing phase.” However, the data of the five consecutive CMJs were not presented. It would be interesting including a table with the means of each jump.”

 I still believe that it is interesting presenting the data of all 5 consecutive jumps. It will help to observe the decrease in the performance as mentioned by the authors. Please, in the next letter of response (not necessarily in the manuscript), provide a table with the average values of first, second, third, fourth, and fifth jumps from all individuals as well as male and female separately.

Round 3

Reviewer 1 Report

Comments and Suggestions for Authors

I suggest presenting all the raw data in the graph, not just the simple mean standard deviation. The graph needs further optimization, font axis annotations, etc.

Author Response

Thank you for the suggestion. The figures have been optimised by completing the axis labels and including the individual data points.

Reviewer 2 Report

Comments and Suggestions for Authors

I'm glad that the suggestions were accepted by authors. After the reviews there is an improvement in the quality of the manuscript. My only minor comment is regarding tables format. I suggest review the journal "instruction for authors" and format accordingly.

Author Response

Thank you for your comments. We have revised the table style accordingly.

Round 4

Reviewer 1 Report

Comments and Suggestions for Authors

All comments have been addressed.